# ASVD: Activation-aware Singular Value Decomposition for Compressing Large Language Models

## Abstract

In this paper, we introduce a new post-training compression paradigm for Large Language Models (LLMs) to facilitate their wider adoption. We delve into LLM weight low-rank decomposition, and find that the challenges of this task stem from ❶ the distribution variance in the LLM activations and ❷ the sensitivity difference among various kinds of layers. To address these issues, we propose a training-free approach called Activation-aware Singular Value Decomposition (ASVD). Specifically, ❶ ASVD manages activation outliers by transforming the weight matrix based on the activation distribution. This transformation allows the outliers in the activation matrix to be absorbed into the transformed weight matrix, thereby enhancing decomposition accuracy. ❷ Additionally, we propose an efficient iterative calibration process to optimize layer-specific decomposition by addressing the varying sensitivity of different LLM layers. In this way, ASVD can compress a network by 10%-30%. Based on the success of the low-rank decomposition of projection matrices in the self-attention module, we further introduce ASVD to compress the KV cache. By reducing the channel dimension of KV activations, memory requirements for KV cache can be largely reduced. ASVD can further achieve 50% KV cache reductions without performance drop in a training-free manner. Code is anonymously available in supplementary materials.

## 1 Introduction

In the realm of Large Language Models (LLMs) compression, various techniques have been extensively explored, including weight quantization [Dettmers et al., 2022], network pruning [Frantar & Alistarh, 2023], and knowledge distillation [Agarwal et al., 2023]. Distinct from these approaches, the paradigm of low-rank matrix decomposition is less explored in LLMs but holds significant promise. Decomposition involves approximating the weight matrices in neural networks with matrices of lower rank, effectively reducing the model size. Given the massive number of parameters in LLMs, low-rank decomposition offers significant potential for memory reduction. Furthermore, low-rank decomposition can complement existing LLM compression techniques by further compressing quantized or pruned models, enhancing overall efficiency [Cheng et al., 2017].

From the perspective of network compression, traditional low-rank decomposition methods typically adhere to a straightforward process: initially training the original model and subsequently fine-tuning the decomposed model [Jaderberg et al., 2014, Khodak et al., 2021, Wang et al., 2021, Hsu et al., 2022]. While this approach is effective, it is resource-intensive and requires the entire training dataset and substantial computational power for end-to-end backpropagation. Applying this method to LLMs would encounter major challenges. Firstly, the training data for LLMs may not always be readily available, often restricted by privacy and commercial considerations. Secondly, the training process for these models is notoriously expensive, both in terms of time and computational resources.

Given these constraints, the concept of "training-free" compression emerges as a more viable approach for LLMs [Zhu et al., 2023]. This approach includes methods like LLM post-training quantization [Dettmers et al., 2022, Yuan et al., 2023] and LLM post-training pruning [Frantar & Alistarh, 2023], which compress LLMs without the need for extensive retraining. These training-free (*i.e., post-training*) methods offer a more practical solution for efficiently compressing LLMs.

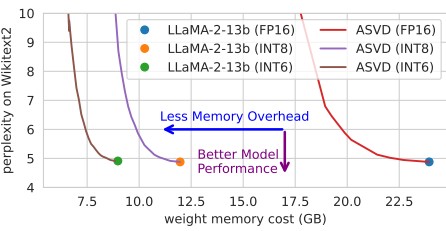

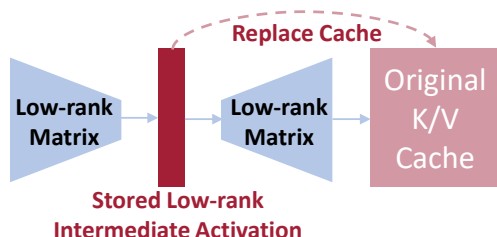

(a) Summarized Performance of `ASVD`.   (b) High-level idea of using `ASVD` to compress KV cache.

Figure 1: **(a)** Our post-training LLM decomposition method is **orthogonal** to existing LLM compression techniques, enabling it to function as a versatile and plug-and-play solution for prevalent compression paradigms, including popular quantization methods. **(b)** By applying low-rank decomposition via `ASVD` to the Key/Value projection matrices, the original high-dimensional **KV cache** can be replaced with a low-dimensional storage.

To realize **LLM low-rank decomposition in a training-free manner**, we conduct an extensive analysis of the baseline methods for LLM decomposition. We first observe that straightforward application of existing low-rank decomposition techniques, which typically necessitate training, turns out ineffective for LLMs [Denton et al., 2014, Lebedev et al., 2014, Sainath et al., 2013, Moczulski et al., 2015, Jaderberg et al., 2014, Khodak et al., 2021, Wang et al., 2021].

Digging into the failures, we reveal two challenges to post-training decomposition for LLMs. ❶ Managing activation distribution in LLMs: This challenge involves addressing outliers in the activations, which can intensify the decomposition error. The importance of handling such outliers in LLMs echoes findings in recent quantization research [Lin et al., 2023, Kim et al., 2023]. These outliers can disproportionately affect the accuracy of matrix approximations, leading to suboptimal compression results. ❷ Balancing layer's decomposition sensitivity: Some layers are more sensitive to the decompostion than others, and decomposing them uniformly can lead to significant performance degradation. The key challenge is to balance the sensitivity of each layer with the efficiency of the whole network's decomposition.

Targeting challenge ❶, we propose the activation-aware decomposition method, where the distribution of activations are considered into the weight decomposition process. Specifically, we transform the values in the weight matrix column-wisely via a scaling matrix. The scaling matrix is designed based on the distribution patterns observed across input activation channels. This adjustment proves particularly beneficial for activation with outliers, allowing the decomposition to allocate enhanced focus to these specific weights. Targeting challenge ❷, we further investigate the varying sensitivity of different LLM layers to decomposition. We find that weights in Multi-Head Attention layers [Vaswani et al., 2017] tend to be more resilient to decomposition compared to those in Multi-Layer Perceptron layers. This sensitivity variability across layers prompts us to develop a method to assign the compression ratio for each layer. `ASVD` assesses each layer's sensitivity to decomposition at different ranks, enabling us to assign a suitable rank for optimal decomposition. Note that this probing assess is very efficient, requiring only a limited sample set for evaluation.

Our experiments reveal that `ASVD` can reduce the rank of the weight matrix by 10% to 90% in different layers, and it can achieve compression of model size 10%-30% in LLaMA models [Touvron et al., 2023a;b]. We also validate `ASVD` is compatible with 4/8-bit weight quantization, which is described in Sect. 4.4.

Importantly, leveraging the successful low-rank decomposition of projection matrices in the self-attention module, we can integrate `ASVD` with KV cache compression. Specifically, by applying `ASVD` to decompose the Key/Value projection matrices, we can derive low-rank intermediate activations that serve as replacements for the KV cache stored in a high-dimension space, as shown in Fig. 1b. This substitution significantly reduces the memory usage of the KV cache, enabling support for larger batch sizes or longer sequence lengths, which are essential for real-world applications [Yuan et al., 2024]. In practice, by replacing the KV cache with intermediate low-rank activations, we can reduce up to 50% of the memory consumption of the KV cache.

## 2 RELATED WORK

**Large Language Model Compression.** The field of model compression for Large Language Models (LLMs) has seen a surge of innovative techniques aimed at mitigating the substantial computation and memory requirements these models demand [Zhu et al., 2023, Yuan et al., 2024]. Various methods

have emerged to address this challenge, each taking a unique approach to reduce the memory footprint of LLMs. These methods primarily fall into three categories: weight quantization [Courbariaux et al., 2015, Dettmers et al., 2022], network pruning [LeCun et al., 1989, Frantar & Alistarh, 2023], and knowledge distillation [Hinton et al., 2015, Agarwal et al., 2023]. For the wide body of research on LLM compression, please refer to [Zhu et al., 2023] for the comprehensive survey. Among these methods, weight quantization has gained significant traction in the context of LLMs due to its effectiveness. However, despite its popularity as a neural network compression technique, low-rank factorization has not been extensively explored in the realm of LLMs. Recognizing this gap, we introduce a novel low-rank decomposition method tailored specifically for decomposing the weight matrices of LLMs in a training-free manner.

**Low-rank Decomposition.** In the realm of low-rank decomposition [Schotthöfer et al., 2022] for neural network compression, existing methods can be broadly classified into two categories: **fixed low rank** and **variable low rank** approaches. Fixed rank methods typically involve decomposing weight matrices of pre-trained networks using techniques like Singular Value Decomposition (SVD) or tensor decomposition, followed by fine-tuning the factorized network [Denton et al., 2014, Lebedev et al., 2014, Sainath et al., 2013, Moczulski et al., 2015]. They also involve constraining weight matrices to maintain a fixed low rank during training [Jaderberg et al., 2014, Khodak et al., 2021, Wang et al., 2021], or constructing layers as linear combinations of layers with varying ranks [Ioannou et al., 2015]. A notable limitation of these methods is the introduction of matrix decomposition rank as a hyperparameter requiring fine-tuning. In contrast, rank-adaptive methods address this limitation by automatically determining and adjusting the low-rank structure. In particular, Kim et al. [2015; 2019] apply heuristics search to pre-determine the decomposition rank, while Wen et al. [2017] learn low-rank weights through a loss function penalizing approximated matrix ranks. Li et al. [2023] use low-rank approximation plus a sparse matrix to compress the weight matrix in transformers.

However, none of these methods have worked in the era of LLMs due to their training-require nature. We propose ASVD, a *post-training* LLM decomposition approach enabling the adaptive determination of SVD ranks to optimize the matrix approximations based on feature activations. To our knowledge, ASVD represents the first attempt to compress the weights of LLMs through decomposition in a training-free manner. Since the introduction of ASVD, there have been subsequent works on training-free LLM decomposition, such as SVD-LLM [Wang et al., 2024] and Palu [Chang et al., 2024]. These follow-up studies underscore the significance and potential of our approach. We hope that our proposed post-training LLM decomposition method can establish a new paradigm for LLM compression, opening up avenues for more efficient and accessible deployment of LLMs. Recently, Lin et al. [2024] highlight a key issue of SVD-based LLM compression methods including ASVD: the full-rank decomposition initially doubles the parameter count of the original model. Consequently, achieving a 90% compression ratio of the original model's parameters requires approximately 55% rank reduction in the decomposed matrices. They observe that more efficient compression can be achieved in layers without intermediate non-linear activation functions, where a 50% rank reduction directly corresponds to a 50% parameter reduction. This paradigm shows more potential of low-rank decomposition for LLM compression.

## 3 METHOD

### 3.1 NAÏVE SVD FOR COMPRESSING WEIGHT MATRIX

Singular Value Decomposition (SVD) can be used to decompose the weights of linear layers, which involves decomposing a weight matrix $\mathbf{W} \in \mathbb{R}^{m \times n}$ into three matrices: $\mathbf{U}$, $\boldsymbol{\Sigma}$, and $\mathbf{V}^T$, such that $\mathbf{W} \approx \mathbf{U}\boldsymbol{\Sigma}\mathbf{V}^T$), where $\boldsymbol{\Sigma}$ is an $m \times n$ diagonal matrix, the diagonal values in $\boldsymbol{\Sigma}$ are the singular values of $\mathbf{W}$, and $\mathbf{U} \in \mathbb{R}^{m \times m}$ and $\mathbf{V} \in \mathbb{R}^{n \times n}$ are corresponding right and left singular vector matrices, respectively [Demmel, 1997].

The SVD compression process for a weight matrix can be summarized in three steps: **Decomposition**: Factorize $\mathbf{W}$ using SVD. **Truncation**: Retain the top $k$ singular values and their corresponding right and left singular vectors. This results in approximated matrices $\mathbf{U}_k$, $\boldsymbol{\Sigma}_k$, and $\mathbf{V}_k^T$, where the right singular vector matrix $\mathbf{U}_k$ is $m \times k$, singular $\boldsymbol{\Sigma}_k$ is $k \times k$, and left singular vector matrix $\mathbf{V}_k^T$ is $k \times n$. The choice of $k$ is critical in balancing the compression ratio and the compressed model's performance. **Reconstruction**: Reconstruct an approximated weight matrix: $\mathbf{W}_k = \mathbf{U}_k\boldsymbol{\Sigma}_k\mathbf{V}_k^T$.

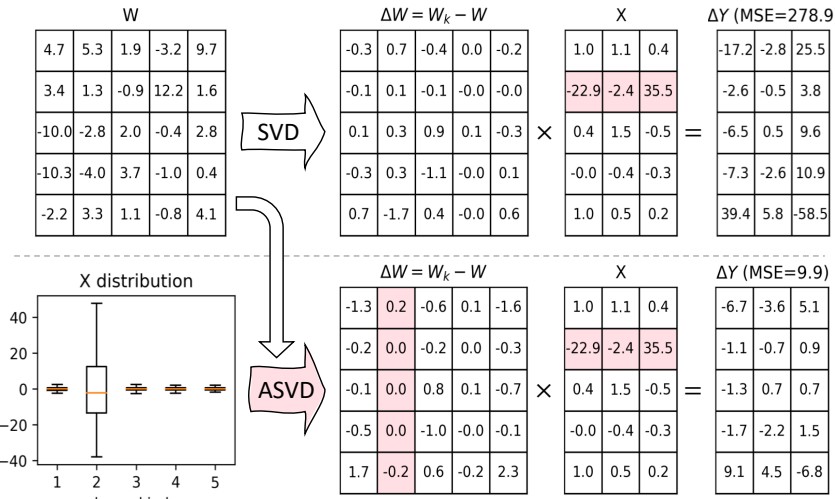

Figure 2: Comparison between SVD and ASVD. Outlier channels in input activations ($\mathbf{X}$) are highlight in red, and ASVD takes these into consideration, which can contribute to a reduction in output error.

## 3.2 CHALLENGES OF COMPRESSING LLMs VIA SVD

Decomposing the large matrices in LLMs (e.g., $4096 \times 4096$ matrices ubiquitous in LLaMA-7b [Touvron et al., 2023a]) into lower ranks presents a viable pathway for model compression. However, straightforward application of existing low-rank decomposition techniques [Denton et al., 2014, Lebedev et al., 2014, Moczulski et al., 2015, Khodak et al., 2021, Wang et al., 2021, Li et al., 2023], which typically necessitate training, proves ineffective for LLMs.

**Challenge 1: Influence of Activation**: This perspective shifts the focus from solely relying on the truncation error $\mathbf{L}_t$, which depends only on the model's weights, to also accounting for the activations. The rationale behind this is the critical role of outliers in activations within LLMs [Lin et al., 2023, Wei et al., 2022, Kim et al., 2023]. Thus, for effective LLM decomposition, our objective optimization becomes:

$$\mathbf{W}_k^\star = \arg\min_{\mathbf{W}_k} \|\mathbf{W}_k\mathbf{X} - \mathbf{W}\mathbf{X}\|_F^2. \tag{1}$$

Here, $\mathbf{X}$ represents the input activations, which are cached from a small calibration set. This set is derived from the pre-training dataset to avoid overfitting to a specific task. Essentially, our objective is to ensure that the output of the decomposed LLM closely mimics the output of the original LLM, rather than merely aligning their weights. This approach prioritizes functional equivalence over structural similarity, recognizing that accurate output replication is more critical for maintaining the model's post-decomposition performance. We define the variation in activations between the compressed matrix $\mathbf{W}_k$ and the original matrix $\mathbf{W}$ as:

$$\Delta\mathbf{Y} = (\mathbf{W}_k - \mathbf{W})\mathbf{X}. \tag{2}$$

To illustrate this concept, we visualize an example of $\mathbf{W}$, $\mathbf{W}_k$ (decomposed by simply SVD), $\mathbf{X}$, and the resulting variation in activations $\Delta\mathbf{Y}$ in Fig. 2 (Top line). This visualization reveals a critical insight: even when the variation in weights $\Delta\mathbf{W} = \mathbf{W} - \mathbf{W}_k$ is relatively minor, the corresponding variation in activations $\Delta\mathbf{Y}$ can be huge. This significant variation in activations is a key factor in why a straightforward SVD-based decomposition approach falls short in effectively decomposing LLMs. The activation variations, despite being derived from input activations of large magnitude (not the weight variations), can lead to considerable changes in the whole model's output, thereby undermining the decomposition's efficacy.

**Challenge 2: Singular Values Variations among Layers**: The distribution of singular values within a matrix is indicative of its sparsity and, by extension, its sensitivity to certain types of information [Kim et al., 2015; 2019, Wen et al., 2017]. In LLMs, there is a notable variation in singular values across different layers. Specifically, some layers exhibit a concentration of large singular values, signifying less sensitivity to weight variation. This characteristic often correlates with these layers being easy to compress. Conversely, other layers in the LLMs display a more uniform distribution of smaller singular values. Such a pattern suggests a balanced contribution from various

singular vector pairs. This variability in the distribution of singular values among layers presents a unique challenge, as it implies that each layer may require a tailored approach to decompose and maintain the overall functionality of the LLM.

These challenges underscore the necessity for innovative approaches specifically designed for the LLM decomposition. Our objective is to achieve efficient compression while circumventing the substantial computational and data demands associated with training-based methods. To address the first challenge, we introduce an Activation-aware SVD mechanism, which is detailed in Section 3.3. This method is designed to mitigate the impact of weight variation on activations. For the second challenge, we propose a Sensitivity-based Truncation Rank Searching mechanism, elaborated in Section 3.4, which adapts to the varying singular value distributions among different layers.

### 3.3 ASVD: ACTIVATION-AWARE SINGULAR VALUE DECOMPOSITION

ASVD is designed to refine the weight matrix $\mathbf{W}$ in LLMs by taking into account the effect of input activation channels. The process comprises the following three steps:

**Transforming the Weight Matrix.** The first step involves transforming the weight matrix $\mathbf{W}$ into an invertible matrix $\mathbf{S}$.

The transform is denoted as $\mathbf{WS}$. Because the matrix $\mathbf{S}$ is invertible, we can have this equation:

$$\mathbf{W} = \mathbf{WSS}^{-1} = (\mathbf{WS})\mathbf{S}^{-1}. \tag{3}$$

**Applying SVD to the Transformed Matrix.** After transforming the weight matrix, the next step is to apply SVD to the transformed matrix $\mathbf{WS}$. The SVD of $\mathbf{WS}$ is expressed as $\mathbf{WS} = \mathbf{U}'\mathbf{\Sigma}'\mathbf{V}'^{T}$. To reduce the elements in these matrices, we truncate them to retain only the top-$k$ singular values. The truncated form of the decomposition is represented as:

$$\mathbf{WS} \approx \mathbf{U}'_k \mathbf{\Sigma}'_k \mathbf{V}'_k{}^{T}. \tag{4}$$

This step ensures that the most significant aspects of the scaled weight matrix are retained. While less critical information, which contributes minimally to the model's output, is discarded.

**Reconstructing the Approximated Weight Matrix.** The final step is to reconstruct an approximation of the original weight matrix. We multiply $\mathbf{V}'_k{}^{T}$ with $\mathbf{S}^{-1}$ to produce a new matrix $\mathbf{V}''_k{}^{T}$:

$$\mathbf{V}''_k{}^{T} = \mathbf{V}'_k{}^{T}\mathbf{S}^{-1}. \tag{5}$$

Note that the matrix $\mathbf{V}''_k{}^{T}$ has the same shape as the matrix $\mathbf{V}'_k{}^{T}$. In this way, the weight matrix can be approximated by:

$$\mathbf{W} = (\mathbf{WS})\mathbf{S}^{-1} \approx (\mathbf{U}'_k \mathbf{\Sigma}'_k \mathbf{V}'_k{}^{T})\mathbf{S}^{-1} = \mathbf{U}'_k \mathbf{\Sigma}'_k \mathbf{V}''_k{}^{T} = \mathbf{W}_k. \tag{6}$$

**Setting the Transform Matrix S.** The transform matrix $\mathbf{S}$ is constructed to adjust $\mathbf{W}$ to better adapt with the activation patterns of the input $\mathbf{X}$. A simple method is to set the transform matrix as a diagonal matrix. The computation of the linear layer can be transformed by:

$$\mathbf{WX} = (\mathbf{WS})(\mathbf{S}^{-1}\mathbf{X}). \tag{7}$$

Each diagonal element in the matrix $\mathbf{S}_{ii}$ transforms the $i$-th input channel of weight as: $(\mathbf{WS})_{:,i} = \mathbf{W}_{:,i}\mathbf{S}_{ii}$. Because $\mathbf{S}^{-1}$ is also a diagonal matrix, the $\mathbf{S}_{ii}^{-1}$ scales the $i$-th channel of the activation as $\mathbf{S}_{ii}^{-1}\mathbf{X}_{i,:}$. This scaling adjusts how each activation channel impacts the weight matrix during the decomposition process. We visualize the impact of the adjustment in Fig.2. We use a small number of corpus sent to the LLM and calculate the absolute mean value of input activation channel. Then we set $\mathbf{S}_{ii}$ according to the absolute mean value of the activations in the $i$-th channel:

$$\mathbf{S}_{ii} := (\frac{1}{n}\sum_{j=1}^{n}|\mathbf{X}_{ij}|)^{\alpha}, \tag{8}$$

where $n$ is the total number of activations for the $i$-th channel and hyper-parameter $\alpha$ provides flexibility to adjust the level of activation sensitivity incorporated into the scaling. This method

focuses on the average magnitude of activation in each channel, capturing the general intensity of activation signals regardless of their positive or negative nature. Since we only need to do the LLM inference several times, this method is very fast.

Another method to set the transform matrix $\mathbf{S}$ to is to optimize the output error introduced by decomposition directly: $\arg\min_{\mathbf{S}} \|\Delta\mathbf{Y}\|_F^2$. Wang et al. [2024] demonstrate that this optimization problem has analytic expression by setting the $\mathbf{S}$ to a lower triangular matrix $\mathbf{L}$, where $\mathbf{L}$ is the Cholesky decomposition of $\mathbf{X}\mathbf{X}^T$:

$$\mathbf{S} := \mathbf{L}, \quad \text{where} \quad \mathbf{L}\mathbf{L}^T = \mathbf{X}\mathbf{X}^T. \tag{9}$$

This method takes an additional step to execute the Cholesky decomposition [Meyer, 2000]. Despite this extra computation, it results in a lower output error $\Delta\mathbf{Y}$.

By designing an invertible transformation matrix $\mathbf{S}$, we can transform the weight matrix $\mathbf{W}$ into a decomposition-friendly matrix $\mathbf{WS}$. This transformation takes into account both input and output activations, making the subsequent decomposition more effective for compression. This is so-called Activation-aware Singular Value Decomposition (ASVD).

### 3.4 Sensitivity-based Truncation Rank Searching

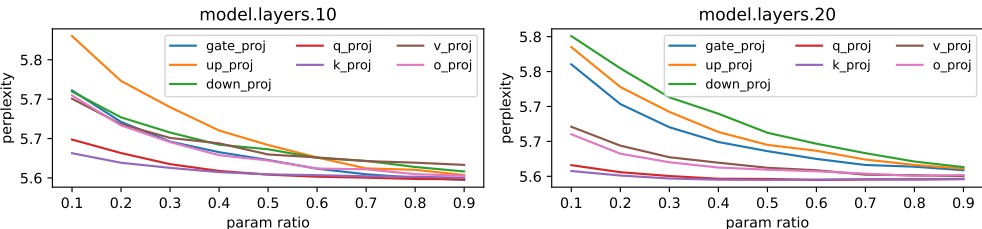

Figure 3: Perplexity across Various Linear Layers and Parameter Ratios on LLaMA-2-7b.

The second challenge arises from the fact that different layers in LLMs exhibit varying degrees of sensitivity to information compression, which is reflected in the distribution of their singular values. Targeting this challenge, we propose the Sensitivity-based Truncation Rank Searching (STRS) method. STRS evaluates the layer sensitivity and decides the best truncation of singular values. In the realm of NLP, perplexity is a key metric for assessing how effectively a language model predicts a sequence of tokens [Brown et al., 2020]. Therefore, we use the reduction in perplexity on the calibration dataset to evaluate the sensitivity of each layer. Similar to post-training compression methods [Dettmers et al., 2022, Frantar et al., 2022, Frantar & Alistarh, 2023], we collect a small number of input token sequences as calibration dataset. Concurrent work by Gao et al. [2024] addresses rank optimization through a differentiable binary masking mechanism. Their method employs regularization to ensure masked ranks maintain consistency with SVD's inherent property of sorted singular values.

The core of the sensitivity evaluation process involves an in-depth exploration of how the neural network reacts to varying levels of truncation. We define a set of potential truncation ratios, denoted as $R = \{0.1, 0.2, \cdots, 0.9\}$. These ratios $r = \frac{km+kn}{mn}$ determine the fraction of the rank $k$ preserved during the SVD truncation for a weight matrix with dimensions $m \times n$. For each linear layer in the LLM, we iterate through these candidate ratios. At each ratio, truncated SVD is applied to the layer's weight matrix, temporarily replacing the original layer in the model with this decomposed version. The model's perplexity is then evaluated on the calibration dataset.

This detailed exploration of sensitivity across various truncation levels provides essential insights into each layer's performance dynamics, informing the optimization and decision-making processes in model compression. As illustrated in Fig. 3, there are noticeable variations in sensitivity among the different layers. Three key observations emerge from this analysis: 1. Inversely Proportional Relationship: lower parameter ratios tend to result in higher perplexity scores. 2. Higher Sensitivity in MLP Layers: MLP layers demonstrate higher sensitivity, indicating where more cautious truncation is necessary. 3. Variable Sensitivity Among Layers: Some layers exhibit relatively lower sensitivity, indicating potential for more aggressive compression.

Assuming the affects of layers are independent, we should set the truncation rank of each layer to minimize the total affect to perplexity under the constraint of parameter size. We propose a binary

search algorithm to search for the best truncation rank. Detailed explanations of algorithm can be found in the Appendix.

### 3.5 ASVD FOR KV CACHE COMPRESSION

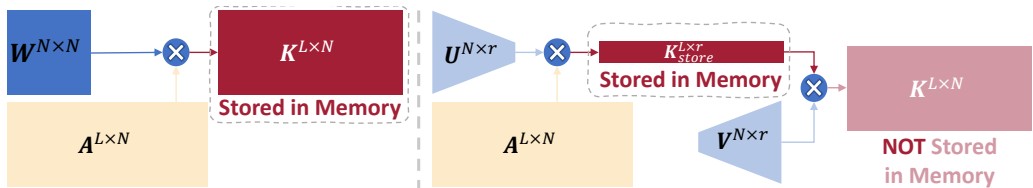

Figure 4: **A demonstration of how ASVD reduces the memory cost of the K cache. (Left)** With long text lengths $L$, the memory required for storing the K cache in a $N$-dimensional space becomes substantial. **(Right)** ASVD decomposes the key projection weight matrix $\mathbf{W}$ into two low-rank matrices $\mathbf{U}$ and $\mathbf{V}$ (see Sec. 3.3). This low-rank structure allows the K representation to be stored in a reduced $r$-dimensional space, where $r \ll N$. Consequently, we only need to save the intermediate K in the $r$ dimension instead of $N$ dimension, saving the K cache $\frac{N}{r}$ times. Note that saving V cache is the same, and when content length $L$ becomes really large (e.g., 1M tokens) or with larger batch size, the KV cache becomes a significant factor in memory cost.

LLM inference with large context lengths can be incredibly resource-intensive, requiring high-end GPUs and, for the largest models, costly multi-GPU setups. Analysis of generative inference with LLMs reveals that, for relatively small batch sizes, the computation is primarily memory-bound [Hooper et al., 2024, Liu et al., 2024]. Given the growing gap between computational speeds and memory speeds, this issue is expected to worsen over time, making it crucial to address the memory bottleneck. Further analysis indicates that the memory bottleneck is strongly correlated with context size. For long sequence lengths, the main contributor to memory consumption is the KV cache storing, so minimizing the KV cache can reduce both memory consumption and bandwidth requirements [Yuan et al., 2024].

As we discussed in Sec.3.3, ASVD decomposes the key and value projection weight matrix $\mathbf{W} \in \mathbb{R}^{N \times N}$ into two low-rank matrices, $\mathbf{U} \in \mathbb{R}^{N \times r}$ and $\mathbf{V} \in \mathbb{R}^{N \times r}$, in a training-free manner, where $N$ is the dimension of K/V embedding space. As shown in Fig.4, replacing the high-rank matrix with two low-rank matrices via ASVD allows us to obtain intermediate activations in low-rank form. These intermediate activations can be stored as a replacement for the original KV cache. In other words, the original KV cache requires storing two $L \times N$ matrices. With ASVD, the new KV cache only needs to store two $L \times r$ matrices. In summary, ASVD can compress the KV cache $\frac{N}{r}$ times. This significant reduction in memory usage for the KV cache enables larger batch sizes or longer sequence lengths, which are critical for real-world applications.

## 4 EXPERIMENTS

In this section, we assess the effectiveness of ASVD by conducting experiments on LLaMA [Touvron et al., 2023a] and LLaMA-2 [Touvron et al., 2023b], and presenting results on various tasks, such as Perplexity in WIKI [Merity et al., 2016] and MMLU [Hendrycks et al., 2020].

### 4.1 SETTINGS

We conducted a comprehensive evaluation of Activation-aware Singular Value Decomposition (ASVD) on two series of Large Language Models (LLMs): LLaMA and LLaMA-2 [Touvron et al., 2023a;b]. Our experiments encompassed models ranging from 7 billion to 13 billion parameters. For each model, we selected a calibration set with 32 samples, and each sample contains 2048 tokens, from the Wikitext dataset to assess the layer-wise sensitivity. We explore two methods to set transform matrix $\mathbf{S}$. The first is the magnitude-based method (Eq.8), which is indicated by ASVD. We set $\alpha$ to 0.5 in our experiments [1]. We also experimented with the Cholesky decomposition method (Eq.9) to set the transform matrix, denoted ASVD+ in our experiments.

---

[1]The exploration of hyper-parameter $\alpha$ can be found in the Appendix.

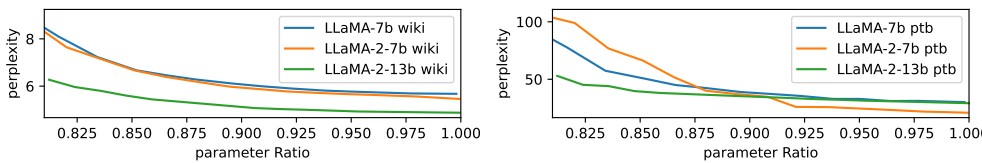

Figure 5: Perplexity trends of `ASVD` compression on LLaMA-2-13b, LLaMA-2-7b and LLaMA-7b.

Table 1: Performance under various compression scenarios. Param ratio indicates the proportion of parameters remaining after decomposition. MMLU results are 0-shot. SVD* means SVD using Sensitivity-based Truncation Rank Searching.

| | | LLaMA-7b | | | LLaMA-2-7b | | | LLaMA-2-13b | | |
|---|---|---|---|---|---|---|---|---|---|---|
| method | param ratio | MMLU | wiki | ptb | MMLU | wiki | ptb | MMLU | wiki | ptb |
| original | 1 | 30.76% | 5.68 | 29.63 | 34.86% | 5.47 | 20.82 | 40.16% | 4.88 | 29.21 |
| SVD | 0.95 | 22.98% | 2800 | 5458 | - | nan | nan | - | nan | nan |
| SVD* | 0.95 | 23.92% | 136.05 | 183.92 | 24.78% | 46.79 | 363.37 | 24.86% | 167.63 | 567.02 |
| SVD* | 0.9 | 23.54% | 698.66 | 262.03 | 24.31% | 114.45 | 27660 | - | nan | nan |
| ASVD | 0.95 | 30.26% | 5.78 | 32.64 | 33.24% | 5.64 | 23.98 | 39.52% | 4.94 | 31.93 |
| ASVD | 0.9 | 29.67% | 6.09 | 37.80 | 32.58% | 5.93 | 32.63 | 40.04% | 5.12 | 34.03 |
| ASVD | 0.85 | 29.70% | 6.80 | 52.11 | 31.57% | 6.74 | 59.84 | 37.95% | 5.54 | 39.32 |
| ASVD | 0.8 | 27.85% | 8.89 | 88.09 | 28.15% | 8.91 | 114.70 | 34.63% | 6.53 | 59.68 |
| ASVD | 0.75 | 24.94% | 14.51 | 212.80 | 25.97% | 18.97 | 432.57 | 28.59% | 8.71 | 110.10 |

## 4.2 PARAMETERS COMPRESSION

Sensitivity-based Truncation Rank Searching (STRS in Sec.3.4) involves setting varying thresholds binary searching process, enabling us to observe the impact of different compression levels on model performance. This approach resulted in a range of compressed networks, each characterized by a unique compression ratio. We evaluated the performance of these compressed networks using perplexity as the primary metric, focusing on two datasets: Wikitext-2 (wiki) and the Penn Treebank (ptb). The results, illustrated in Fig.5, reveal several key insights: (1) As the parameter ratio decreases, there is a corresponding increase in perplexity. (2) A plateau region is observed when the parameter ratio exceeds 0.9. In this range, `ASVD` predominantly decompresses the less sensitive layers, resulting in minimal impact on prediction accuracy. (3) Below a parameter ratio [2] of 0.85, there is a rapid increase in perplexity, indicating that the more sensitive layers are being decompressed to a lower truncation rank, adversely affecting the model's performance.

We also present a detailed analysis of the performance of compressed networks at various parameter ratios. Table 1 displays the performance metrics for two LLaMA models, LLaMA-7b and LLaMA-2-7b, under several compression scenarios. These metrics include MMLU zero-shot evaluation, perplexity on the Wikitext dataset (wiki), and perplexity on the Penn Treebank dataset (ptb). Our observations reveal significant performance variations based on the parameter ratio and the compression method used. Specifically, the table highlights the performance of each model when using standard SVD, SVD with binary search for truncation ranks (SVD*), and `ASVD` at different parameter ratios ranging from 0.75 to 0.95.

We compare `ASVD` and `ASVD`+[3] with SVD-LLM [Wang et al., 2024]. The results in Table 2 show that `ASVD`+ can improve the performance of `ASVD`, especially when the compression ratio is high. `ASVD`+ also outperforms the SVD-LLM method, particularly when the compression ratio is less than 30%. This is because SVD-LLM does not consider the layer-wise differences. In contrast, our method uses Sensitivity-based Truncation Rank Searching to set each layer with a different compression ratio. However, when the compression ratio is larger than 30%, all of these methods can significantly improve the perplexity of the LLMs.

Table 2: The perplexity on wikitext2 of SVD-LLM, ASVD and ASVD+. In this table, we take the setting of SVD-LLM that the lm head is not taken into consideration to compute the param ratio.

| | LLama-2-7b | | | | LLama-2-13b | | |
|---|---|---|---|---|---|---|---|
| Param ratio | SVD-LLM | ASVD | ASVD+ | Param ratio | SVD-LLM | ASVD | ASVD+ |
| 0.95 | 6.93 | 5.64 | 5.56 | 0.95 | 5.70 | 4.94 | 4.93 |
| 0.9 | 7.27 | 5.93 | 5.74 | 0.9 | 5.94 | 5.12 | 5.03 |
| 0.85 | 7.76 | 6.74 | 6.10 | 0.85 | 6.24 | 5.54 | 5.26 |
| 0.8 | 8.38 | 8.91 | 6.86 | 0.8 | 6.66 | 6.53 | 5.77 |
| 0.75 | 9.30 | 18.97 | 8.38 | 0.75 | 7.22 | 8.71 | 6.54 |
| 0.7 | 10.67 | 159.21 | 10.62 | 0.7 | 8.00 | 20.82 | 7.82 |
| 0.65 | 12.82 | 1034.59 | 13.87 | 0.65 | 9.10 | 53.30 | 9.84 |
| 0.6 | 16.14 | 730.60 | 19.12 | 0.6 | 10.78 | 133.88 | 13.18 |

Table 3: Performance under different KV cache compression ratio.

| | | KV cache ratio | | | | | | | | |
|---|---|---|---|---|---|---|---|---|---|---|
| model | dataset | 1(original) | 0.9 | 0.8 | 0.7 | 0.6 | 0.5 | 0.4 | 0.3 | 0.2 |
| LLaMA-2-7b | wiki | 5.47 | 5.46 | 5.48 | 5.50 | 5.55 | 5.67 | 5.94 | 6.55 | 8.71 |
| | ptb | 20.82 | 21.04 | 21.52 | 21.66 | 21.91 | 22.16 | 24.33 | 26.89 | 38.72 |
| LLaMA-2-13b | wiki | 4.88 | 4.89 | 4.90 | 4.91 | 4.92 | 4.96 | 5.08 | 5.33 | 6.06 |
| | ptb | 29.21 | 29.64 | 29.95 | 30.21 | 30.99 | 31.69 | 34.03 | 36.61 | 47.24 |

## 4.3 KV CACHE COMPRESSION

We evaluate the KV Cache compression by using ASVD to decompose k projection and v projection in transformer [Vaswani et al., 2017]. Table 3 summarizes the results, showing the perplexities on the wikitext2 and Penn Treebank datasets for different KV cache compression ratios. It is evident that the perplexity values remain stable when the KV cache ratio is above 40%. When the ratio is lower than 40%, the performance of the network is decreased. These observations suggest that ASVD is effective to compress the KV cache without negatively impacting the model.

## 4.4 INTEGRATING ASVD WITH QUANTIZATION

This section investigates the compatibility of ASVD with quantization techniques for compressing LLMs. We explore the integration of ASVD with different quantization methods. Simple quantization methods include Round-To-Nearest (RTN) and 4-bit NormalFloat (NF4) [Dettmers et al., 2023]. The advanced LLM quantization method is Activation-aware Weight Quantization (AWQ) [Lin et al.,

---

[2] parameter ratio 0.85 means compress the model size by 15%.

[3] ASVD+ refer to ASVD with whitening method for obtaining the transformation matrix in Eq.9

Table 4: Combining weight quantization with ASVD. Param ratio indicates the proportion of parameters remaining after ASVD, with 1 implying no decomposition.

| | LLaMA-2-7b | | | | | LLaMA-2-13b | | | | |
|---|---|---|---|---|---|---|---|---|---|---|
| param ratio | FP16 | INT8 (RTN) | INT8 (AWQ) | NF4 | INT4 (AWQ) | FP16 | INT8 (RTN) | INT8 (AWQ) | NF4 | INT4 (AWQ) |
| | | | | | wiki | | | | | |
| 1 | 5.47 | 5.48 | 5.45 | 5.65 | 5.59 | 4.88 | 4.88 | 4.88 | 4.98 | 4.97 |
| 0.95 | 5.64 | 5.64 | 5.56 | 5.83 | 5.82 | 4.94 | 4.95 | 4.97 | 5.08 | 5.18 |
| 0.9 | 5.93 | 5.94 | 5.82 | 6.2 | 6.21 | 5.12 | 5.11 | 5.15 | 5.31 | 5.43 |
| 0.85 | 6.74 | 6.73 | 6.51 | 7.43 | 7.18 | 5.54 | 5.56 | 5.59 | 5.9 | 5.96 |
| | | | | | ptb | | | | | |
| 1 | 20.82 | 20.82 | 20.93 | 22.7 | 21.50 | 29.15 | 29.12 | 29.29 | 30.31 | 30.47 |
| 0.95 | 23.98 | 23.95 | 25.47 | 35.91 | 27.79 | 31.93 | 31.67 | 30.19 | 33.89 | 31.21 |
| 0.9 | 32.63 | 32.19 | 37.11 | 40.82 | 39.31 | 34.03 | 33.64 | 35.47 | 34.93 | 38.95 |
| 0.85 | 59.84 | 63.76 | 84.52 | 427.59 | 95.85 | 39.32 | 40.02 | 43.01 | 44.49 | 50.56 |

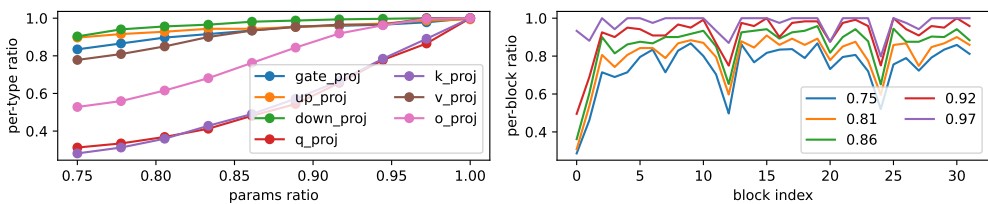

Figure 6: Per-type parameters ratio and per-block parameters ratio on LLaMA-2-7b after `ASVD` compression.

2023]. Note that our study focuses on establishing the orthogonal property of ASVD to these basic quantization methods. Future work could extend this investigation to more advanced quantization techniques and other LLM compression approaches. Our experimental framework involves two stages. Firstly, we apply `ASVD` to decompose the network. Subsequently, we quantize the decomposed weights.

Table 4 summarizes the results of our experiments on LLaMA-2-7b, LLaMA-2-13b models [Touvron et al., 2023b]. The following observations were made: **8-bit Weight Quantization:** The results indicate that 8-bit quantization has a negligible impact on model performance, both for the original and the ASVD-compressed networks. **4-bit weight Quantization:** Upon quantizing the network into NF4 and INT4(AWQ), a further deterioration in prediction accuracy is observed. When param ratio is greater than 0.9, the performance decline attributed to quantization is approximately consistent with that of the non-decomposed network. We observe that the performance degradation of LLaMA-2-13b is less than that of LLaMA-2-7b, indicating that the larger model is more robust to compression. In summary, the findings suggest that `ASVD` is compatible with weight quantization techniques.

### 4.5 DECOMPOSED NETWORK ANALYSIS

We conduct a detailed analysis of the decomposed network. Figure 6 presents the per-type parameters ratio and per-block parameters ratio. Observing the plot, we note that parameters in the MLP components (gate projection, up projection, and down projection) exhibit minimal compression. In MHA, the V projection layer experiences relatively small compression, whereas q projection and k projection can be significantly compressed, indicating redundancy in these components. Turning our attention to the per-block compression ratio, we find that the first layer can undergo substantial compression. In contrast, the compression ratios for the other layers, except for two middle layers, show similar compression rates.

This computation ratio can be expressed as the ratio of $C_k$ to $C$, which is equivalent to the parameter ratio:

$$\frac{C_k}{C} = \frac{km + kn}{nm} \tag{10}$$

Remarkably, this computation ratio mirrors the weight number compression ratio, highlighting the efficient use of computational resources achieved through `ASVD`. In summary, `ASVD` can not only reduce the weight storage and weight transferring overheads in LLM deployment but also reduce the computation required by LLM inference.

## 5 CONCLUSION

This study presents a training-free approach to compressing Large Language Models (LLMs). We propose Activation-aware Singular Value Decomposition (`ASVD`) and Sensitivity-based Truncation Rank Searching (STRS), effectively address the challenges posed by activation outliers and varying layer sensitivities. These techniques enable more accurate and efficient decomposition, reducing memory usage and computational demands while maintaining model performance. The successful integration of `ASVD` into KV cache compression further underscores its potential for broad applicability and substantial impact in real-world scenarios.

## 6 REPRODUCIBILITY STATEMENT

We have submitted the code for the experiments as part of the supplementary material. The code is anonymous and self-contained and includes detailed instructions to facilitate the replication of our experiments and findings. We also plan to publicly release the code, data, pretrained models, and any additional resources needed for the community to fully reproduce our work.

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

# A APPENDIX

## A.1 IMPACT STATEMENTS AND LIMITATIONS

In this study, we propose a technique that improves the efficiency of Large Language Models (LLMs), making them more accessible. This approach helps to democratize LLMs by lowering deployment costs and hardware barriers, facilitating their use in edge computing. However, it does not mitigate the potential misuse of LLMs by malicious actors.

Despite the remarkable achievements of the ASVD method in compressing large language models (LLMs), several limitations persist. One limitation arises when ASVD is combined with quantization techniques, which can lead to a decline in model performance. While 8-bit weight quantization has minimal effects on both original and ASVD-compressed networks, switching to 4-bit quantization can result in a slight decrease in predictive accuracy. Additionally, ASVD faces difficulties in compressing multi-layer perceptron (MLP) in LLMs, as these layers typically contain more parameters than self-attention mechanisms, resulting in increased computational burdens due to their high-dimensional feature mappings. Although ASVD effectively compresses the weights in multi-head attention (MHA) with fewer parameters, it struggles with MLP. Furthermore, the need to evaluate the sensitivity of each layer requires a forward propagation step to calculate perplexity, demanding significant computational resources.

## A.2 RELEASE SAFEGUARDS

While ASVD itself does not release new pretrained models, the compression capabilities it provides could enable easier sharing and deployment of powerful models that have risks of misuse. To mitigate risks of misuse, we have implemented access control. Users must agree to terms prohibiting unethical applications.

## A.3 INFERENCE COST WITH DECOMPOSED LLMS

Regarding the computational aspect, let's consider the input matrix $\mathbf{X} \in \mathbb{R}^{n \times t}$ and the weight matrix $\mathbf{W} \in \mathbb{R}^{m \times n}$. In the original linear layer, the matrix multiplication is represented as $\mathbf{Y} = \mathbf{W}\mathbf{X}$. The number of Multiply-Accumulate (MAC) operations, denoted as $C$, in the original linear layer can be computed as: $C = tmn$. After the ASVD decomposition, the matrix multiplication transforms into $\mathbf{Y} \approx \mathbf{U}'_k \mathbf{\Sigma}'_k \mathbf{V}''_k \mathbf{X}$. We can fuse the $\mathbf{\Sigma}_k$ into $\mathbf{U}'_k$ and $\mathbf{V}''_k$. Then we have:

$$\mathbf{Y} \approx \mathbf{U}'_k \mathbf{\Sigma}'_k \mathbf{V}''_k \mathbf{X} \tag{11}$$

$$= (\mathbf{U}'_k \sqrt{\mathbf{\Sigma}'_k})(\sqrt{\mathbf{\Sigma}'_k} \mathbf{V}''_k)\mathbf{X} \tag{12}$$

$$= \mathbf{A}\mathbf{B}\mathbf{X} \tag{13}$$

To analyze the computational efficiency, we calculate the MAC operations, denoted as $C_k$, for this decomposed form. The computation for $C_k$ is given by: $C_k = tkm + tkn$

This computation ratio can be expressed as the ratio of $C_k$ to $C$, which is equivalent to the parameter ratio:

$$\frac{C_k}{C} = \frac{km + kn}{nm} \tag{14}$$

Remarkably, this computation ratio mirrors the weight number compression ratio, highlighting the efficient use of computational resources achieved through ASVD. In summary, ASVD can not only reduce the weight storage and weight transferring overheads in LLM deployment but also reduce the computation required by LLM inference.

## A.4 BINARY SEARCH FOR TRUNCATION RANKS

We have the option to employ either a performance target or parameters target for our search. In the case of a performance target, our objective is to identify the truncation rank configuration that ensures the compressed network attains the desired performance, such as achieving a specific perplexity. Alternatively, in the pursuit of a parameters target, our goal is to identify the truncation ranks that result in the network attaining the specified target parameters.

---

**Algorithm 1:** Binary Search for Truncation Ranks (parameters target)

---

**Input:** List of tuples (layer, truncation rank, sensitivity) and parameters target
**Output:** Optimal truncation rank configuration for each layer
Sort the list by sensitivity in ascending order
Initialize pointers: $p_L = 0$, $p_H = $ length of list $- 1$
$p_M = \left\lfloor \frac{p_L + p_H}{2} \right\rfloor$
**while** $p_L \neq p_H$ **do**
    **for** each layer in the list **do**
        Initialize $r = \infty$
        **for** each tuple in the list starting from $p_M$ to the end **do**
            **if** tuple's layer is the same as the current layer **then**
                $r = \min(r, \text{tuple's truncation rank})$
            **end if**
        **end for**
        **if** $r = \infty$ **then**
            Do not modify the truncation rank for the layer
        **else**
            Set the truncation rank for the layer to $r$
        **end if**
    **end for**
    Calculate the parameters after compression
    **if** parameters $\leq$ parameters target **then**
        $p_H = p_M$
    **else**
        $p_L = p_M + 1$
    **end if**
    Update $p_M = \left\lfloor \frac{p_L + p_H}{2} \right\rfloor$
**end while**

---

The algorithm of performance target: Initially, the low pointer ($p_L$) is positioned at the start of the list, while the high pointer ($p_H$) is set at the list's end. The middle pointer ($p_M$), as the name suggests, is placed midway between $p_L$ and $p_H$, calculated as $p_M = \left\lfloor \frac{p_L + p_H}{2} \right\rfloor$. During each iteration of the binary search, we adjust the truncation rank for each layer. Specifically, for a given layer, its truncation rank is set to the smallest rank found to the right of the middle pointer ($p_M$) in our list.

Following this adjustment, we evaluate the network's performance using the updated configuration on a calibration dataset. The primary metric for assessment is perplexity. Should the perplexity fall within or below a pre-established threshold, we move the high pointer ($p_H$) to the middle position ($p_M$). This action indicates our search for a configuration with a potentially lower rank that still adheres to performance standards. Conversely, if the perplexity exceeds our maximum acceptable threshold, we shift the low pointer ($p_L$) to ($p_M + 1$). This adjustment signifies the need to increase the truncation ranks to maintain or enhance performance levels. The binary searching will converge to an optimal configuration of truncation ranks for each layer that balances compression ratio and perplexity.

The algorithm of parameters target is shown in Algorithm 1. It doesn't need calibration dataset.

## A.5 DIFFERENCE WITH TENSORGPT.

In the content of LLM compression via decomposition, the most related work is the concurrent TensorGPT Xu et al. [2023], Zhu et al. [2023], in which the embedding layer of LLMs is compressed through Tensor-Train Decomposition (TTD) Oseledets [2011] in order to store large embeddings in a low-rank tensor format, with much fewer parameters. However, there are several differences between those two methods: (1) Unlike TensorGPT which focuses solely on the token embedding matrix, ASVDaims to compress the entire weight spectrum of LLMs. This holistic approach addresses a more critical aspect of LLM compression, as highlighted in recent studies Lin et al. [2023], Kim et al. [2023]; (2) From the perspective of low-rank decomposition categorization, our method can realize

the low-rank decomposition in a rank-adaptive manner, contrasting with the fixed or predetermined ranks used in TensorGPT.

### A.6 EMPIRICAL COMPARISON WITH FWSVD

We also compare ASVD with FWSVD Hsu et al. [2022], which uses Fisher information to weigh the importance of parameters affecting the model prediction. Note that FWSVD is training-required. As shown in Table 5, our method can outperform FWSVD comprehensively.

Table 5: Comparing with FWSVD on LLaMA-7b. FWSVD* denotes Fisher information weighted SVD.

| param ratio | | 0.95 | 0.9 | 0.85 | 0.8 |
|---|---|---|---|---|---|
| LLaMA-7b | | | | | |
| FWSVD+STRS | wiki | 5.86 | 6.32 | 7.48 | 10.70 |
| ASVD | | 5.78 | 6.09 | 6.80 | 8.89 |
| FWSVD+STRS | ptb | 34.33 | 38.05 | 58.75 | 125.80 |
| ASVD | | 32.64 | 37.80 | 52.11 | 88.09 |
| LLaMA-2-7b | | | | | |
| FWSVD+STRS | wiki | 5.59 | 6.12 | 8.01 | 13.07 |
| ASVD | | 5.64 | 5.93 | 6.74 | 8.91 |
| FWSVD+STRS | ptb | 25.06 | 36.58 | 105.53 | 222.03 |
| ASVD | | 23.98 | 32.63 | 59.84 | 114.70 |

### A.7 HYPER-PARAMETERS EXPLORATION

Table 6: Perplexity on Wikitext2 for exploring hyper-parameters on OPT-125m.

| $\alpha$ | 0.1 | 0.25 | 0.5 | 1 | 2 |
|---|---|---|---|---|---|
| SVD+STRS | | | 103.39 | | |
| ASVD abs mean | 47.54 | 37.12 | **36.89** | 41.53 | 43.81 |
| ASVD abs max | 52.63 | 47.17 | 40.14 | 41.94 | 52.55 |

In our study, we initiate an exploration of hyper-parameters in `ASVD`, focusing on the activation channel significance metric and the control factor $\alpha$. This exploration is conducted on OPT-125m, a relatively small network that facilitates rapid evaluation.

We rigorously explored the control factor $\alpha$ at various settings: 0.1, 0.25, 0.5, 1, and 2. This exploration aimed to understand how varying $\alpha$ influences the performance and parameter efficiency of the network. Additionally, we investigated two methods for quantifying activation significance: Absolute Mean Value of Input Activation and Absolute Maximum Value of Input Activation. These methods are crucial in determining the most effective approach for activation channel significance evaluation. We set a target parameters ratio of 0.9. Utilizing the binary search approach for truncation ranks, we report the perplexity on Wikitext2 test set after compression. The results of our experiments are summarized in Table 6.

From the data presented in the table, we observe that both activation-aware methods show superior performance compared to standard SVD+STRS. We also notice that Lower and higher values of $\alpha$ (0.1 and 2) exhibit lower performance, while mid-range values (0.5) lead to better performance, and the Absolute Mean Value method consistently outperforms the Absolute Max Value method. Therefore, based on our observations, we chose $\alpha = 0.5$ and the Absolute Mean Value method for setting the transform matrix $\mathbf{S}$ in the ASVD process in the following experiments.

### A.8 ABSORBING SINGULAR VALUES

After we decompose a matrix via `ASVD`, we can represent the weight matrix as a product of three matrices, i.e., $\mathbf{W} \approx \mathbf{U}_k \mathbf{\Sigma}_k \mathbf{V}_k^T$. Thanks to the diagonal nature of matrix $\mathbf{\Sigma}_k$, we can further optimize the inference process. Specifically, we can efficiently absorb the singular values in $\mathbf{\Sigma}_k$ into the

Table 7: Perplexity on Wikitext-2 under different absorbing strategies after `ASVD` on OPT-125m.

| param ratio | weight quant | absorbed by UV | absorbed by U | absorbed by V |
|---|---|---|---|---|
| 0.9 | INT6 | **37.58** | 39.67 | 40.62 |
| 0.85 | INT6 | **60.44** | 64.19 | 61.02 |

matrices $U_k$ and $V_k^T$. We achieve this fusion using the following strategy: $\mathbf{A}_k = \mathbf{U}_k\sqrt{\Sigma_k}$ and $\mathbf{B}_k = \sqrt{\Sigma_k}\mathbf{V}_k^T$. Consequently, we obtain a more computationally efficient matrix operation:

$$\mathbf{Y} = \mathbf{W}\mathbf{X} \approx \mathbf{A}_k(\mathbf{B}_k\mathbf{X}) \tag{15}$$

Compared to the methods of fusing the singular values $\Sigma_k$ solely into either $\mathbf{U}$ or $\mathbf{V}$ matrices, our proposed fusion technique offers significant advantages in terms of weight quantization, as demonstrated in Table 7. Our approach involves evenly distributing the singular values from the diagonal matrix $\Sigma_k$ into both $\mathbf{U}_k$ and $\mathbf{V}_k^T$ matrices. This ensures a more uniform distribution of $\mathbf{A}_k$ and $\mathbf{B}_k$, leading to a reduction in the disparity across different channels and reducing the quantization error.

