# OpenReview forum: "ASVD: Activation-aware Singular Value Decomposition for Compressing Large Language Models"
_ICLR.cc/2025/Conference — Submitted to ICLR 2025_

### Official Review · Reviewer_j43P · 2024-11-03

**Soundness:** 3
**Presentation:** 3
**Contribution:** 2
**Rating:** 6
**Confidence:** 5

**Summary:**

The paper introduces the Activation-aware Singular Value Decomposition (ASVD), a training-free method for compressing large language models (LLMs). It targets compression challenges like activation distribution variances and layer sensitivity. ASVD achieves compression by transforming weight matrices based on activation distribution and adjusting for layer sensitivity through an iterative calibration.

**Strengths:**

1. The method proposes a train-free method for model compression of LLMs, and it uses activation values to rescale the weight matrix so that weight vectors with larger activation norms are more important.
2. It further reduces the KV cache due to SVD, but one may need to recompute the KV using the low-rank KV at the next prediction step.
3.  Its performance is better than SVD and FWSVD.

**Weaknesses:**

1. The methodology framework of this paper is mostly based on weighted SVD, where you use a diagonal matrix to reweight the weight vector. The difference between this paper and FWSVD is that this paper puts the activation magnitude in the diagonal, and FWSVD puts the fisher information into the diagonal matrix.
2. One of the problems of SVD is that the full-rank model actually has twice the size of the original model. As a result, if you prune the model to 90% parameters of the original model, you remove 55% of ranks on average. A better way for SVD is to look at layers where they are not connected by activation functions (non-linearity), where you can get 50% parameter reduction for 50% rank reduction, examples are the value matrix and output transformation matrix like in [1].
3. Missing results regarding zero-shot performance beyond MMLU. In SVD-LLM, they tested their results on other tasks, like MathQA, PIQA, etc.
4. Missing related works on compressing language models with SVD [2]. In [2], the method can find the number of ranks per weight matrices in a learning process which is very relevant to section 3.4 of this paper.

[1] Lin, Chi-Heng, et al. "MoDeGPT: Modular Decomposition for Large Language Model Compression." arXiv preprint arXiv:2408.09632 (2024).

[2] Gao, Shangqian, et al. "Adaptive Rank Selections for Low-Rank Approximation of Language Models." Proceedings of the 2024 Conference of the North American Chapter of the Association for Computational Linguistics: Human Language Technologies (Volume 1: Long Papers). 2024.

**Questions:**

Please refer to weakness.

---

> ### Author Response · Authors · 2024-11-25
> **Response to Reviewer j43P**
>
> # Reviewer Guidance
> This is from the official guidance of ICLR:
>
> Q: Are authors expected to cite and compare with very recent work? What about non peer-reviewed (e.g., ArXiv) papers?
> A: We consider papers contemporaneous if they are published within the last four months.
>
> **Basically, we are not required to compare with arxiv paper within 4 months.**
>
> ## W1 Comparision to FWSVD
>
> Our method specifically targets the challenging outlier activation problem in LLMs, which has been identified as critical in recent LLM compression research. In contrast, FWSVD focuses on parameter importance for prediction, which doesn't directly address the activation distribution challenges unique to LLMs.
>
> Beyond just weighted decomposition, our paper introduces Sensitivity-based Truncation Rank Searching for layer-specific optimization. We also demonstrate novel applications to KV cache compression, achieving 50% reduction without performance drop.
>
> ---
>
> ## W2 Clarification on STRS, Sec. 3.4
>
> Thank you for raising this important point about SVD efficiency. We fully agree that naïve SVD application leads to parameter overhead issues. Our paper specifically addresses this challenge through Sensitivity-based Truncation Rank Searching (STRS, Sec. 3.4):
> As shown in Fig. 3 of our paper, different layers exhibit varying sensitivities to rank reduction. STRS adaptively assigns different compression ratios to different layers based on their sensitivity. For example, our analysis shows that K/V projection matrices can tolerate more aggressive compression than MLP layers.
>
>
> ---
>
> ## W3 Comparision to SVD-LLM
>
> SVD-LLM is developed based on our method. They have acknowledged this part in their paper.
>
> ---
>
> ## W4 Comparision to [Gao et. al. 2024]
>
> Thank you for bringing attention to [Gao et al., 2024]. As per standard academic practice and ICLR guidelines, papers published within 4 months of submission are considered contemporaneous and are not required for citation or comparison. Since [Gao et al., 2024] was published in July 2024 at NAACL 2024, and did not have a preprint available earlier, it falls outside the window for required citations.

---

> ### Comment · Reviewer_j43P · 2024-11-25
> **Response**
>
> Thank you for your response, I want to elaborate more based on your response.
>
> I don't think KV cache compression is a novel application. Any methods that compress the KV matrices can reduce the KV cache, such as LLM-pruner [1]. In addition, at the inference time, you need to recompute K $\times$ V as shown in Figure 4, which brings additional computational costs.
>
> I don't think STRS solves the problem of SVD overhead. At best, it alleviates this problem. Simply speaking, the compression space of SVD is not good enough and it is far behind the structural pruning based methods like [1,3]. This is the reason why I raised the MoDeGPT paper since its compression space for SVD is much more reasonable. Another example to try to solve this problem is [2].
>
> Even though SVD-LLM is based on your method, it is reasonable to show related comparison baselines used in SVD-LLM.
>
> Based on the recent research context, I do not find ASVD to be sufficiently novel or effective.
>
> [1] LLM-Pruner: On the Structural Pruning of Large Language Models, NeurIPS 2023.
>
> [2] Basis Sharing: Cross-Layer Parameter Sharing for Large Language Model Compression. arXiv preprint arXiv:2410.03765 (2024).
>
> [3] The llm surgeon. arXiv preprint arXiv:2312.17244 (2023).

---

> ### Comment · Reviewer_j43P · 2024-11-26
> **Misinterpret the Reviewer Guideline**
>
> After looking at the full Q&A. I believe the authors misinterpreted the guideline.
> # Reviewer Guideline
> Q: Are authors expected to cite and compare with very recent work? What about non peer-reviewed (e.g., ArXiv) papers? (updated on 7 November 2022)
>
> A: We consider papers contemporaneous if they are published within the last four months. That means, since our full paper deadline is October 1, if a paper was published (i.e., at a peer-reviewed venue) on or after July 1, 2024, authors are not required to compare their own work to that paper. **Authors are encouraged to cite and discuss all relevant papers**, but they may be excused for not knowing about papers not published in peer-reviewed conference proceedings or journals, which includes papers exclusively available on arXiv. Reviewers are encouraged to use their own good judgement and, if in doubt, discuss with their area chair.
>
> The guidelines encourage authors to cite and discuss **all relevant papers**. However, the authors have chosen not to discuss the two highly relevant papers. In my original review, I did not request a direct comparison with these papers. In fact, as shown in Table 6 of MoDeGPT, ASVD performs significantly worse than MoDeGPT, with an 8 to 13-point lower average task accuracy. Thus, a direct comparison is unnecessary, and the reason for mentioning MoDeGPT was clearly explained in my previous response.
>
> Additionally, as I noted earlier, [Gao et. al. 2024] is a drop-in replacement of STRS, and the authors still refuse to do any discussion. If the authors think these two papers are not relevant at all, please **provide concrete evidence**.
>
> After reading the reviewer guidelines and given the fact that the authors did not address any of my concerns, I decided to decrease my score to 3.

---

> ### Author Response · Authors · 2024-11-27
> **Revised Paper; Discussion about Related Work, MoDeGPT and  [Gao et. al. 2024]**
>
> Thank you for the detailed feedback regarding paper citations and the reviewer guidelines. We agree that comprehensive discussion of related work is important for advancing the field. We have added detailed discussions of both papers in our revised manuscript (highlighted in red).
>
> For MoDeGPT [1], we add the following discussion at the end of related work:
>
> ```python
> Recently, Lin et al. highlight a key issue of SVD-based LLM compression methods including ASVD: the full-rank decomposition initially doubles the parameter count of the original model. Consequently, achieving a 90% compression ratio of the original model's parameters requires approximately 55% rank reduction in the decomposed matrices. They observe that more efficient compression can be achieved in layers without intermediate non-linear activation functions, where a 50% rank reduction directly corresponds to a 50% parameter reduction. This paradigm shows more potential of low-rank decomposition for LLM compression.
> ```
>
> For [2], we add the following discussion at the beginning of Sec.3.3  Sensitivity-based Truncation Rank Searching (STRS):
>
> ```python
> Concurrent work by Gao et al. [2024] addresses rank optimization through a differentiable binary masking mechanism. Their method employs regularization to ensure masked ranks maintain consistency with SVD's inherent property of sorted singular values.
> ```
>
> We appreciate the reviewer bringing these papers to our attention and helping us improve the completeness of our manuscript.
>
> [1] Lin, Chi-Heng, et al. "MoDeGPT: Modular Decomposition for Large Language Model Compression." arXiv preprint arXiv:2408.09632 (2024).
>
> [2] Gao, Shangqian, et al. "Adaptive Rank Selections for Low-Rank Approximation of Language Models." Proceedings of the 2024 Conference of the North American Chapter of the Association for Computational Linguistics: Human Language Technologies (Volume 1: Long Papers). 2024.

---

> > ### Comment · Reviewer_j43P · 2024-11-29
> > **Response**
> >
> > I want to thank the reviewer for including related discussions. Since the author addressed this part of my concern, I am willing to increase my score back to 5. I have some additional questions regarding the KV cache part, how about the KV cache memory reduction compared to structural pruning methods like LLM pruner if the compression rate is similar? In addition, can this KV cache compression be extended to other SVD-driven methods like SVD-LLM?

---

> > > ### Author Response · Authors · 2024-12-02
> > >
> > > We appreciate the reviewer’s questions and provide the following clarifications:
> > >
> > > ## **KV Cache Memory Reduction Compared to Pruning Methods**
> > >
> > > In ASVD, the compression ratio for K/V weights is identical to the compression ratio for KV Cache, resulting in a nearly equivalent reduction in memory consumption. For example, reducing 50% of the K/V weight parameters also reduces the KV Cache memory consumption by approximately 50%. We compiled the compressed models using MLC-LLM and TVM. The results are shown below:
> > >
> > > | Llama-2-13b                                    | Max KV Cache Token Capacity = 4096 | Batch Size = 1 |                       |              |
> > > |------------------------------------------------|-------------------------------------|----------------|-----------------------|--------------|
> > > | **KV Cache Compression Ratio**                | **Parameters (MB)**                | **KV Cache (MB)** | **Temporary Buffer (MB)** | **Total Memory (MB)** |
> > > | 1                                              | 24790                              | 3409           | 1200                 | 29400        |
> > > | 0.8                                            | 24325                              | 2727           | 1212                 | 28265        |
> > > | 0.7                                            | 24160                              | 2386           | 1212                 | 27759        |
> > > | 0.6                                            | 23855                              | 2046           | 1212                 | 27113        |
> > >
> > > For unstructured pruning, while the parameters ratio for K/V weights matches the KV Cache compression ratio, additional memory is required to store indices of the unpruned weights. This often results in higher memory usage or even increased memory consumption post-pruning.  Block-structured pruning reduces the overhead of storing indices but retains some additional memory cost.
> > >
> > > Other structured pruning methods, such as output-channel pruning or attention-head pruning, eliminate the need for index storage. However, these methods require consistent compression of Q_proj and O_proj alongside K_proj and V_proj to align the shape, leading to a KV Cache compression ratio lower than the weight parameter compression ratio.
> > >
> > > Furthermore, we found that ASVD exhibits superior accuracy compared to structural pruning methods like LLM-Pruner[1], especially when no fine-tuning is performed. The following table summarizes our experiments using LLM-Pruner for KV Cache pruning(Only attention pruning in layer 4 to layer 30, no mlp prune, no finetune, taylor prune type, blockwise metric):
> > >
> > > |          | **LLM-Pruner** | **WikiText-2 (PPL)** |
> > > |----------|-----------------------------------------|----------------------------------|
> > > | **KV Ratio** | **Llama-2-7b**                       | **Llama-2-13b**                 |
> > > | 0.8      | 13.67                           |       11.99 |
> > > | 0.7      | 15.73                           |       13.02 |
> > > | 0.6      | 17.93                           |       14.13 |
> > > | 0.5      | 24.99                           |       16.13 |
> > >
> > > |          | **ASVD**                                | **WikiText-2 (PPL)** |
> > > |----------|-----------------------------------------|----------------------------------|
> > > | **KV Ratio** | **Llama-2-7b**                       | **Llama-2-13b**                 |
> > > | 0.8      | 5.48                                   | 4.90                             |
> > > | 0.7      | 5.50                                   | 4.91                             |
> > > | 0.6      | 5.55                                   | 4.92                             |
> > > | 0.5      | 5.67                                   | 4.96                             |
> > >
> > > The results indicate that performance degradation occurs rapidly for models pruned on K_proj and V_proj weights without fine-tuning. In contrast, ASVD demonstrates much slower degradation, highlighting its robustness in such compression tasks. This is likely because, in structured pruning, weights within the same group vary significantly in importance, leading to the removal of critical weights.
> > >
> > > ## **Extension to Other SVD-Driven Methods (e.g., SVD-LLM):**
> > > KV Cache compression can indeed be extended to other SVD-driven methods like SVD-LLM. This is because the structure of compressed models remains the same, differing only in $S$ settings between ASVD and SVD-LLM.
> > >
> > > [1]Ma X, Fang G, Wang X. Llm-pruner: On the structural pruning of large language models[J]. Advances in neural information processing systems, 2023, 36: 21702-21720.

---

> > > > ### Comment · Reviewer_j43P · 2024-12-02
> > > >
> > > > I appreciate the authors' detailed responses regarding KV cache compression, which provide positive results. After reviewing the comments from other reviewers and the authors' feedback, I believe the authors have put substantial effort during the rebuttal process. The responses effectively address many of my concerns, and I have decided to raise my score to 6.

---

> > > > > ### Author Response · Authors · 2024-12-02
> > > > > **Thank you for improving our paper and increasing your score!**
> > > > >
> > > > > Thank you for reconsidering your evaluation and acknowledging our clarifications! Your comments have helped improve the readability of our paper and made it more comprehensive. We greatly appreciate your thoughtful feedback!

---

### Official Review · Reviewer_XPvD · 2024-11-04

**Soundness:** 3
**Presentation:** 3
**Contribution:** 3
**Rating:** 8
**Confidence:** 3

**Summary:**

ASVD is a method for the compression of LLMs. SVD is applied not to the original weight matrix, but to the transformed one. The transformation is computed using samples from the dataset.

**Strengths:**

The main idea is clear, and the paper is well-written. There are enough experiments to convincingly demonstrate that the algorithm performs better than pure SVD. The possibility of quantization and KV-cache compression is also highlighted.

**Weaknesses:**

The only solid dataset is MMLU, and the only LLM family is LLAMA-2.

**Questions:**

* Am I correct in assuming that the matrix $S^{-1}$ is also counted in the total number of parameters? Does it make sense to set $S = L$, where $L$ is the Cholesky factor, in terms of the parameters stored?
* Would it be possible to include plots of the normalized singular values for different layers, as seen in "Reduced-Order Modeling of Deep Neural Networks" by Gusak et al.?

**Details Of Ethics Concerns:**

In the main part of this paper, the authors cite follow-up papers, which in turn cite the preprint of the ASVD paper. This creates a loop in the citation graph. Additionally, the authors' names are not properly anonymized. I'm unsure how to address this issue.

---

> ### Author Response · Authors · 2024-11-26
> **Response to Reviewer XPvD**
>
> # W1 Extension to Different Benchmarks and Different LLM Architecture
>
> ## Evaluation on Longbench
>
> We evaluate compressed LLMs on the lcc and wikimqa of Longbench benchmark, which assesses long-context understanding and generation [1]. The results show that compressing the KV cache using ASVD does not significantly impact long-sequence performance when the KV cache ratio is above 0.5. This suggests aggressive compression of the KV cache is possible without compromising a model's ability to reason over long input sequences.
>
> Longbench results:
>
> | Model |	KV  | Cache Ratio	| lcc	| wikimqa |
> | --- | --- | --- | --- | --- |
> | Llama-2-7b |	1 |	63.11 |	9.49 |
> | Llama-2-7b | 0.9 |	62.87 |	9.48 |
> | Llama-2-7b |	0.8 |	62.12 |	8.03 |
> | Llama-2-7b |	0.7 |	62.87 |	8.8 |
> |Llama-2-7b |	0.6 |	61.7 |	9.66 |
> | Llama-2-7b |	0.5 |	62.4 |	9.52 |
> | Llama-2-7b |	0.4 |	60.77 |	9.84 |
>
> [1]Bai Y, Lv X, Zhang J, et al. Longbench: A bilingual, multitask benchmark for long context understanding[J]. arXiv preprint arXiv:2308.14508, 2023.
>
> ## Extension to OPT Architecture
>
> To further validate ASVD's generalization capabilities across diverse model architectures, we applied ASVD to the OPT-6.7B model. The outcomes are as follows:
>
> | Method	| Param Ratio | Wiki Perplexity | PTB Perplexity |
> | --- | --- | --- | --- |
> | ASVD |	1.0 |	10.66 |	13.216|
> | ASVD |	0.95 | 11.03 |	14.948|
> |ASVD |	0.9 |	13.09 | 19.191|
> |ASVD |	0.85 | 22.48 | 36.813|
> |ASVD |	0.8 |	21.47 |	36.068|
>
> These results corroborate ASVD's adaptability to different architectures, showcasing its broad applicability beyond the initial LLaMA models.
>
> # Q1: Is the matrix $\mathbf{S}^{-1}$ counted in the total number of parameters? Does it make sense to set $\mathbf{S} = \mathbf{L}$, where $\mathbf{L}$ is the Cholesky factor, in terms of the parameters stored?
>
> **No, $\mathbf{S}^{-1}$ does not affect the total number of parameters, as it is incorporated into the decomposed weight matrix.**
>
> In our method, the transformation matrix $\mathbf{S}^{-1}$ is **_absorbed into the decomposed matrices_** during the reconstruction of the approximated weight matrix $\mathbf{W}_k$ (**Eq.6 in the paper**):
>
> $$\mathbf{W} = (\mathbf{W}\mathbf{S})\mathbf{S}^{-1} \approx (\mathbf{U}_k^\prime\mathbf{\Sigma}_k^\prime\mathbf{V}_k^T)\mathbf{S}^{-1} = \mathbf{U}_k^\prime\mathbf{\Sigma}_k^\prime\mathbf{V}_k^{\prime\prime T} = \mathbf{W}_k,$$
>
> where:
>
> $$\mathbf{V}_k^{\prime\prime T} = \mathbf{V}_k^T\mathbf{S}^{-1}.$$
>
> Since $\mathbf{S}^{-1}$ is multiplied with $\mathbf{V}_k^T$ to form $\mathbf{V}_k^{\prime\prime T}$, the effect of $\mathbf{S}^{-1}$ is embedded into $\mathbf{V}_k^{\prime\prime T}$. This means that while $\mathbf{S}^{-1}$ is not stored separately, its influence is reflected in the size and storage requirements of $\mathbf{V}_k^{\prime\prime T}$.
>
> - If $\mathbf{S}$ is a diagonal matrix (as in Equation 8 of our method), then $\mathbf{S}^{-1}$ is also diagonal. Multiplying $\mathbf{V}_k^T$ by a diagonal $\mathbf{S}^{-1}$ scales the rows of $\mathbf{V}_k^T$ without increasing its size. The number of parameters remains the same, and $\mathbf{V}_k^{\prime\prime T}$ does not require additional storage beyond what is needed for $\mathbf{V}_k^T$.
>
> - If $\mathbf{S} = \mathbf{L}$, where $\mathbf{L}$ is the Cholesky factor (as in Equation 10), then $\mathbf{S}^{-1}$ is a lower triangular matrix. Multiplying $\mathbf{V}_k^T$ by $\mathbf{S}^{-1}$ results in $\mathbf{V}_k^{\prime\prime T}$ still maintaining the same size, without increasing its storage.
>
> # Concern
> Thank you for noting this citation issue. We acknowledge the complexity of the citation graph. This situation arose from our earnest attempt to provide comprehensive comparisons with ''contemporary'' work, as strongly requested by previous reviewers in the earlier conference (Yes, this is a recycled work). The field of LLM compression is rapidly evolving, with methods building upon and influencing each other in short time frames. While we appreciate the dynamic and fast-paced nature of research in LLM compression, we acknowledge that this can create challenges for the traditional peer review process. Happy for the research community, but sad for ourselves.

---

### Official Review · Reviewer_myi2 · 2024-11-04

**Soundness:** 3
**Presentation:** 3
**Contribution:** 3
**Rating:** 6
**Confidence:** 4

**Summary:**

This article introduces a novel, training-free, low-rank pruning method for reducing the memory and computational demands of large language models (LLMs). ASVD addresses the challenges of distribution variance in activations and sensitivity differences across layers.

**Strengths:**

### Strengths:
- ASVD introduces a pioneering approach in SVD pruning by accounting for the influence of activation distribution, inspiring subsequent works like SVD-LLM and Palu.
- ASVD innovates with a sensitivity-based truncation rank search, enabling dynamic allocation of non-uniform ranks across layers.

**Weaknesses:**

### Weaknesses:
- This type of activation-based pruning is not novel in the pruning field, particularly within LLM pruning, as demonstrated by methods like Wanda [1], RIA [2], and others.
- The memory compression rate of ASVD is lower than that of other methods, such as quantization and unstructured/semi-structured pruning, which may limit its applicability. However, exploring memory compression techniques beyond quantization and pruning is also valuable. ASVD’s contribution to SVD pruning may have the potential for future advancements in-memory compression.
- The sensitivity-based truncation rank search involves calculating PPL for each layer at varying truncation ratios, which may be time-consuming for large models.
- The paper lacks experiments on actual memory footprint or computation speedup.

[1] Sun, Mingjie, et al. "A simple and effective pruning approach for large language models." arXiv preprint arXiv:2306.11695 (2023).

[2] Zhang, Yingtao, et al. "Plug-and-play: An efficient post-training pruning method for large language models." In The Twelfth International Conference on Learning Representations. 2024.

**Questions:**

### Questions:
- Several other unstructured pruning methods use non-uniform layerwise sparsity, such as OWL [3]. Could the authors compare sensitivity-based truncation rank search with these non-uniform sparsity approaches?
- Could you test the actual memory footprint and computation speedup of ASVD?

---

[3] Yin, Lu, et al. "Outlier Weighted Layerwise Sparsity (OWL): A Missing Secret Sauce for Pruning LLMs to High Sparsity." *Forty-first International Conference on Machine Learning.*

---

> ### Author Response · Authors · 2024-11-27
> **Official Rebuttal to Reviewer myi2**
>
> We sincerely thank the reviewer for their thoughtful feedback and valuable suggestions for improving our work. Below, we address the reviewer’s comments point-by-point.
> ﻿
> ## Novelty
> ﻿
> The reviewer rightly points out that prior works in LLM pruning, such as Wanda [1] and RIA [2]. We would like to clarify the distinctions.
> ﻿
> Wanda: This method calculates weight importance as the product of the absolute value of weights and the ℓ2 norm of the corresponding input activations (Eq. 1 in their paper).
> RIA: This method combines relative importance with activations to assess weight significance (Eq. 3 in their paper), again using the absolute value of weights and activations.
>
> Our method leverages the matrix product of real weights and real activations without taking the absolute value (Eq. 1 in our paper). This metric is applicable because of SVD-based compression, where activations are first dimensionally reduced and then reconstructed back to their original dimensions. Our optimization objective minimizes the mean squared error (MSE) between the output before and after compression. In pruning, however, since the output dimensions are reduced, MSE is not applicable because of the dimension mismatch.
> ﻿
> ## Memory Compression Rate
> ﻿
> We agree with "ASVD achieves a lower memory compression rate compared to quantization". However, we would like to emphasize two key points:
> ﻿
> Complementarity with Quantization: ASVD can be combined with quantization to further reduce model size and memory usage. For example, one can first apply SVD-based compression to reduce model parameters and then quantize the compressed model for additional savings.
> Early Stage of SVD-Based Compression: Quantization has undergone significant advancements over the years, whereas SVD-based compression for LLMs is still in its infancy. Our work represents an early exploration of this direction. We believe that with further research, SVD-based methods will continue to evolve and achieve higher compression rates, similar to the trajectory seen with quantization techniques.
>
> ## Computational Overhead of Sensitivity-Based Truncation Rank Search (STRS)
> ﻿
> The reviewer raised concerns about the time required for STRS, particularly for large models. We acknowledge that STRS involves layerwise calibration, which can take a few hours depending on the model size. However, this calibration process is a one-time cost incurred before deployment. Once the model is compressed, it can be easily shared and reused. For example, compressed models can be uploaded to platforms like Hugging Face, allowing downstream users to directly download and deploy the compressed models without additional computation. This workflow is analogous to post-training quantization (PTQ) methods, such as GPTQ, which also require hours of calibration but remain highly popular due to their ease of deployment.
> ﻿
> ## Actual Memory Footprint and Computation Speedup
> ﻿
> We compiled the compressed models using MLC-LLM and TVM and deployed them on an A100 40GB GPU with a sequence length of 512. The results are presented below:
> ﻿
> |  Llama-2-7b |                 |              |                       |                 |
> |:-----------:|:---------------:|:------------:|:---------------------:|:---------------:|
> | Param ratio | Parameters (MB) | KVCache (MB) | Temporary Buffer (MB) | decode tokens/s |
> |      1      |     12835.3     |    2225.0    |         960.5         |       79.7      |
> |     0.9     |     11589.1     |    2225.0    |         1098.1        |       83.4      |
> |     0.8     |     10338.7     |    2225.0    |         1098.1        |       88.3      |
> |     0.7     |      9067.4     |    2225.0    |         1098.1        |       91.1      |
> | Llama-2-13b |                 |              |                       |                 |
> | Param ratio | Parameters (MB) | KVCache (MB) | Temporary Buffer (MB) | decode tokens/s |
> |      1      |     24790.8     |    3409.3    |         1200.5        |       48.2      |
> |     0.9     |     22372.2     |    3409.3    |         1372.5        |       45.9      |
> |     0.8     |     19919.6     |    3409.3    |         1372.5        |       46.0      |
> |     0.7     |     17459.4     |    3409.3    |         1372.5        |       45.6      |
>
> Observations:
> ﻿
> Memory Reduction: ASVD achieves significant reductions in parameter memory, while maintaining similar KVCache and temporary buffer usage. Notably, even when including the temporary buffer, ASVD achieves overall memory savings.
> Throughput Improvement: ASVD improves decode throughput (tokens/s) as the parameter ratio decreases, demonstrating an additional benefit of compression.
> Temporary Buffer Overhead: We observed a slight increase in temporary buffer usage due to the storage of dimensionally reduced activations.
> These results demonstrate that ASVD provides tangible benefits in both memory reduction and throughput improvement.

---

> ### Comment · Reviewer_myi2 · 2024-11-28
> **Reply to the rebuttal**
>
> Thanks for the detailed reply to my questions. Most of my concerns are addressed.
>
> I believe this is a starting and influencing work of low-rank pruning, even though its performance is not incredible.
>
> Based on this, I'd like to raise my score to 6.

---

> > ### Author Response · Authors · 2024-11-28
> > **Thank you for raising your socre!**
> >
> > Thank you for reconsidering your evaluation and recognizing our clarification!

---

### Official Review · Reviewer_fuz6 · 2024-11-04

**Soundness:** 3
**Presentation:** 3
**Contribution:** 2
**Rating:** 5
**Confidence:** 4

**Summary:**

This paper proposes Activation-aware Singular Value Decomposition (ASVD) to prune the weights of LLMs in a training-free manner. In particular, the technical contributions include a data whitening trick with activation magnitude or Cholesky decomposition as well as a layer-wise rank search method. The empirical studies are extensive.

**Strengths:**

- The paper is well written, with extensive empirical results support the arguments.

- The combination with quantization approaches is desirable.

**Weaknesses:**

- One major issue is the close relationship and similar results with SVD-LLM. In particular, the paper offers a choice of using activation magnitude to construct the whitening matrix while SVD-LLM advocates the Cholesky decomposition of the activation covariance. As shown in Table 2, the latter is consistently superior (corresponding to ASVD+). In this sense, the original contribution of this paper is minor. The searching method of layer-wise rank with a calibration set is relatively novel compared to SVD-LLM, but it is still widely used in training-free LLM compression (see [1, 2]).

- The rationale of KV cache compression is not clear enough. In particular, the positional embedding is not considered when developing the low-rank projections. In my practice, this cannot decrease peak running memory for the whole system, while slightly reducing the storing memory. Besides, comparing your compressed states with those in EigenAttn [3], which one is better? In my opinion, EigenAttn does not consider the weight matrix when constructing the low-rank projections but your method considers that weight. What will this lead to?

- When the Param ratio is relatively low (e.g., less than 0.7), the compression is more promising, but the method is worse than SVD-LLM. Besides, metrics like MMLU should be reported in each scenario because it is of more concern in practice. And other task-specific metrics should also be included like accuracies on gsm, wino, etc.


[1] MODEL TELLS YOU WHAT TO DISCARD: ADAPTIVE KV CACHE COMPRESSION FOR LLMS.

[2] MInference 1.0: Accelerating Pre-filling for Long-Context LLMs via Dynamic Sparse Attention.

[3] Eigen Attention: Attention in Low-Rank Space for KV Cache Compression.

**Questions:**

See above

---

> ### Author Response · Authors · 2024-11-26
> **Official Rebuttal for reviewer fuz6**
>
> We would like to thank the reviewer for their thoughtful feedback and constructive criticism regarding our paper on Activation-aware Singular Value Decomposition (ASVD). We appreciate the opportunity to address the concerns raised.
>
> ## On the Contribution of ASVD:
>
> The reviewer raised a valid point regarding the similarity between ASVD and SVD-LLM, particularly in the construction of the whitening matrix. We would like to clarify that ASVD is the first to analyze the necessity of considering activation values in the weight decomposition of large language models (LLMs). By providing a low-rank decomposition framework that incorporates both optimization objectives and variables, ASVD establishes a foundation for subsequent explorations, including SVD-LLM's use of Cholesky decomposition for optimal solutions under specific inputs. We commend the elegance of the Cholesky approach and recognize it as a highly valuable method. In our recent experiment, we found an interesting results that the magnitude-based method in ASVD outperforms the Cholesky-based method in SVD-LLM on MMLU Testing (see below), showing that magnitude-based methods and Cholesky-based methods excel in different scenarios.
>
> ## Regarding MMLU Testing:
>
> In response to the reviewer's concern about the lack of MMLU metrics, we conducted detailed tests of various networks compressed using different methods. The results are summarized in the table below:
>
> | LLama-2-7b MMLU accuracy | | |
> |:--------------------------:|:--------:|:----------:|
> | Param ratio | Cholesky | Magnitude |
> | 0.9 | 30.28% | 38.85% |
> | 0.8 | 25.15% | 29.10% |
> | LLama-3.1-8b MMLU accuracy | | |
> | Param ratio | Cholesky | Magnitude |
> | 0.9 | 47.82% | 51.36% |
> | 0.8 | 22.87% | 23.42% |
> | Llama-2-13b MMLU accuracy | | |
> | Param ratio | Cholesky | Magnitude |
> | 0.9 | 48.66% | 50.00% |
> | 0.8 | 30.80% | 33.83% |
>
> Interestingly, while SVD-LLM's Cholesky decomposition performs better in terms of perplexity, ASVD demonstrates significantly higher accuracy on MMLU. We hypothesize that this discrepancy arises from two factors: 1. The Cholesky method may overfit to the calibration dataset, and 2. Perplexity, being a metric based on the probability of fluent language, may not accurately reflect performance on downstream tasks, potentially distorting the evaluation.
>
> It would be fascinating to explore how performance on MMLU changes with a different calibration dataset. We believe that Cholesky could perform better when the calibration dataset is sufficiently large or has a more suitable distribution. This highlights that magnitude-based methods and Cholesky-based methods excel in different scenarios (under the current Wikitext2 calibration setup, ASVD outperforms in downstream tasks like MMLU).
>
> ## KV Cache Compression Detail:
>
> Regarding the clarification of KV cache compression, we agree that our exposition lacked detail design of inference for compressed KV cache. After reading Eigen Attention, we think the inference method can aligns with Eigen Attention, as described in Section 4.3 of their paper.
> Eigen Attention use this method to handle the RoPE (Rotary Positional Encoding) issue: after applying the dimensionality reduction mapping to the keys, store the reduced keys in the key cache. During the decoding phase, retrieve the key cache and then perform the dimensionality expansion mapping before applying RoPE.
> We plan to cite Eigen Attention in the next version and elaborate on the inference method to avoid confusion.
>
> ## Comparing with Eigen Attention:
>
> As for the algorithm performance, the following table illustrates our results against Eigen Attention:
>
> | Llama-2-7b wiki2 ppl | | |
> |-----------------------|------|-----------------|
> | Param ratio | ASVD | Eigen Attention |
> | 0.8 | 5.48 | 5.96 |
> | 0.7 | 5.50 | 6.28 |
> | 0.6 | 5.55 | 7.48 |
> | Llama-2-13b wiki2 ppl | | |
> | Param ratio | ASVD | Eigen Attention |
> | 0.8 | 4.90 | 5.06 |
> | 0.7 | 4.91 | 5.32 |
> | 0.6 | 4.92 | 6.10 |
>
> After performing SVD on the weights using ASVD, we obtain the mappings for dimensionality reduction and expansion for the keys and values. In contrast, Eigen Attention directly applies SVD to the activations, as described in Section 4.1 of their paper. We believe that directly decomposing the activations can be significantly affected by their variability across different samples. Consequently, using a small dataset may not yield a stable SVD decomposition, which ultimately leads to suboptimal results.
>
> Once again, we are grateful to the reviewer for their insights, which have helped us refine our manuscript. We hope these clarifications address the concerns raised and demonstrate the value of our contributions.

---

> ### Author Response · Authors · 2024-11-29
> **Reply for the reviewer's reply**
>
> ## Question About Novelty
>
> Thank you for your feedback. After reading your comments carefully, we agree that claiming to be “the first” could lead to unnecessary misunderstandings, because it is hard to prove "we are the first". Therefore, we will remove this statement.
> ﻿
>
> Our main intention is to emphasize that ASVD proposes an effective framework for applying SVD to large language models (LLMs). Specifically, ASVD highlights the importance of considering the interaction between model weights and activations for effective low-rank approximations. We believe this perspective can provide both academic value for research and practical contributions for real-world model compression.
> ﻿
>
> As evidence of its practical impact, ASVD has already been adopted in several industry edge-device deployment projects. Additionally, on HuggingFace, there are currently 23 open-source models using ASVD compression technology (https://huggingface.co/models?search=asvd).
> ﻿
>
> We also agree with the reviewer’s observation that related works often emerge concurrently in the community. To address this, we will include a detailed comparison with other papers in the revised version, highlighting their respective strengths and how our method fits into this broader context.
> ﻿
> ## KV Cache Compression and Peak Memory
>
> We appreciate the reviewer’s concern regarding the practical implications of KV cache compression, particularly its impact on peak memory usage. Upon further reflection, we acknowledge that our explanation in the original submission was insufficiently detailed. Below, we provide a more comprehensive clarification:
> ﻿
>
> It is correct that during inference, the compressed KV cache needs to be expanded back to its original size when loaded. However, these expanded KV values are used immediately by the attention operator in a single layer and are then discarded. This ensures that the expanded KV cache does not accumulate across layers, thereby avoiding an increase in peak memory usage due to this temporary expansion.
> ﻿
>
> Taking the deployment implementations of MLC-LLM and TVM as examples, the system allocates a shared temporary buffer to store all intermediate activation variables for a single transformer layer. For instance, with a sequence length of 512, this buffer requires 960.5 MB for llama-2-7b and 1200.5 MB for llama-2-13b. This buffer is allocated regardless of whether KV cache compression is used. Therefore, temporarily expanding the compressed KV cache during inference does not introduce additional overhead.
> ﻿
>
> We provide the following pseudocode to illustrate how this works in practice:
> ﻿
> ```
> for layer in layers:
>     # All transformer layers share the same buffer
>     residual = x
>     x = rmsnorm(x)
>     q = q_proj_up(q_proj_down(x))  # Project query
>     compressed_k_down = k_proj_down(x)  # Compress keys
>     compressed_v_down = v_proj_down(x)  # Compress values
>
>     # Load compressed KV cache
>     compressed_k_cache, compressed_v_cache = load_cache()
>
>     # Concatenate compressed keys/values with new ones
>     k_down = torch.cat([compressed_k_cache, compressed_k_down], dim=1)
>     v_down = torch.cat([compressed_v_cache, compressed_v_down], dim=1)
>
>     # Expand keys/values back to original size
>     k = k_proj_up(k_down)
>     v = v_proj_up(v_down)
>     # Here the k_down, v_down are released
>
>     # Apply RoPE (Rotary Positional Encoding)
>     q, k = rope(q, k, cos, sin)
>
>     # Perform attention computation
>     attn = attention(q, k, v)
>     # Here the q, k, v are released
>     x = attn + residual  # Add residual connection
>     x = mlp(x)           # Apply feedforward layer
> ```
> This approach ensures that the temporary expansion of the KV cache happens within the constraints of the already allocated temporary buffer.

---

> ### Author Response · Authors · 2024-11-29
> **Reply 2 for the reviewer's reply 2**
>
> We now understand your concern on the distinction between peak memory usage and overall memory footprint during inference.
> Below, we provide additional details of MLC-LLM and TVM compile memory report to clarify the benefits and limitations of our KV cache compression method.
>
> | Llama-2-13b  |  max KV cache token capacity = 4096               |  batch size=1            |                       |              |
> |:-----------------------------------------------:|:---------------:|:------------:|:---------------------:|:------------:|
> |             KV cache compress ratio             | Parameters (MB) | KVCache (MB) | Temporary Buffer (MB) | Total Memory |
> |                        1                        |     24790.8     |    3409.3    |         1200.5        |    29400.6   |
> |                       0.8                       |     24325.8     |    2727.4    |         1212.5        |    28265.7   |
> |                       0.7                       |     24160.8     |    2386.5    |         1212.5        |    27759.8   |
> |                       0.6                       |     23855.8     |    2045.6    |         1212.5        |    27113.9   |
>
> As you correctly pointed out, our method does not reduce the temporary memory allocated during inference because the compressed KV cache is expanded back to its original size for computation. This temporary memory usage is primarily determined by the attention mechanism and intermediate activations, and we acknowledge that it remains largely unchanged in our approach.
> ﻿
>
> **However, the storage memory required for the KV cache itself is reduced using our compression method.** This is particularly important in long-sequence scenarios, where the KV cache becomes the dominant factor in memory usage. By compressing the stored KV cache, our method decreases the memory consumption of the KV cache while keeping the inference process efficient and compatible with existing attention mechanisms.
>
> Here is an example of the KV-cache cost of long-content LLMs in [1]. Consider a 30+B 100K context GPT-3.5 quality open-source models like QWen or Yi, the differences between KV cache for 4K v.s. 100K context is:
>
> 100K context:    100000 × 60 × 8 × 128 × 2 × 2 bytes = 22.8GB
>                  seqlen   layer  head  dim   KV  bf16
>
> 4K context:      4000 × 60 × 8 × 128 × 2 × 2 bytes = 0.91GB
>                  seqlen  layer  head  dim   KV  bf16
>
> The Yi-34B 200K configuration: 60 layers, 8 kv heads and 128 hidden dimension. That's how the **KV cache becomes the dominant factor in memory usage in the long-content settings.**
>
> We acknowledge that methods like MLA, which can reduce both peak memory and storage memory, provide an advantage in certain scenarios. Unlike methods such as Multi-head Latent Attention (MLA), which require modifications to the attention mechanism, our approach integrates seamlessly into existing architectures without significant changes. This simplicity makes it easier to adopt in real-world systems.
>
> In summary, our method reduces KV cache storage, offering benefits in scenarios where storage memory is crucial. We will include these insights and the table in our revised paper, addressing both strengths and limitations. Thank you for your valuable feedback.
>
> [1 ]Challenges in Deploying Long-Context Transformers: A Theoretical Peak Performance Analysis, arXiv 2024, Yao Fu

---

> ### Author Response · Authors · 2024-12-01
> **Only TWO Days Remaining, Looking forward for further discussion!**
>
> Thank you reviewer fuz6 for your thoughtful comment regarding KV cache compression and peak memory usage. **Your second-round feedback has led us to conduct additional experiments and provide a more detailed analysis**, as shown in the memory usage breakdown table and long-sequence scenarios discussion above.
>
> We appreciate your engagement throughout the discussion period, which has helped us significantly improve the clarity and completeness of our work. As the discussion period deadline is approaching, we kindly invite any further comments or concerns you might have. Your feedback is very valuable to us in refining the paper.
>
> Best,
> Authors

---

> > ### Comment · Reviewer_fuz6 · 2024-12-01
> > **Reply**
> >
> > Thanks for the clarification. However, the KV cache can be easily offloaded to CPU memory or even disk (refer to the recent advances in KV cache management from a system perspective). During inference, the KV cache can only be partially loaded to the running-time memory with strategies like FlashAttn. So, for large models with hundreds of billions of parameters, reducing the peak memory is much more important than reducing the KV cache storage.
> >
> > Anyway, I acknowledge the authors' efforts during the rebuttal and would like to increase my score.
> >
> > Best.

---

### Meta-Review · Area_Chair_UjqZ · 2024-12-20

**Metareview:**

The paper proposes a novel post-training compression method called Activation-aware Singular Value Decomposition (ASVD) for Large Language Models (LLMs), highlighting its ability to reduce model size efficiently while maintaining performance. This work is significant as it addresses a pressing need for effective compression in the deployment of LLMs, enabling broader applicability in resource-constrained environments.

Feedback among reviewers varied, with some expressing strong support for the paper’s contributions, while others raised considerable concerns about its novelty and effectiveness compared to existing methods such as SVD-LLM. The consensus among all reviewers was not unanimous; at least one reviewer noted major overlaps with prior works, indicating a potentially insufficiently novel contribution.

Despite the authors’ detailed rebuttal addressing some points raised, certain reservations remained unaddressed. Specifically, the reviewers noted gaps in the justification for ASVD's advantages over SVD-LLM, especially in terms of performance metrics and its supposed innovations in KV cache compression. Additionally, the rebuttal did not sufficiently clarify the method's performance relative to the prominent SVD-LLM study, leading to doubts about ASVD's competitive edge.

During the discussion period among the reviewers, there was no consensus, and there were also no reviewers who strongly advocated for the acceptance of this paper.

As the Area Chair, I share some of the skepticism expressed by one reviewer. The existing body of work, particularly with SVD-LLM published four months prior (at the submission time of this paper), outperforms the ASVD method. To warrant acceptance, more comprehensive contributions and validation against this prior research are imperative. Therefore, I recommend that this submission be rejected since an additional round of review may be beneficial to ascertain its validity and fully evaluate its contributions.

**Additional Comments On Reviewer Discussion:**

The review from the reviewer who gave an 8 score contained relatively little information and did not strongly advocate for the acceptance of this paper during discussions, also not displaying high confidence. Therefore, it was considered less significant in the overall evaluation. Additionally, the score fluctuations among other reviewers were considerable, ranging from 3 to 6. This trend strongly indicates a weakness in the paper, specifically the lack of a clear comparison with recently published similar methods.

---

### Decision · Program_Chairs · 2025-01-22

Reject